# Alpha-Synuclein Interaction with UBL3 Is Upregulated by Microsomal Glutathione S-Transferase 3, Leading to Increased Extracellular Transport of the Alpha-Synuclein under Oxidative Stress

**DOI:** 10.3390/ijms25137353

**Published:** 2024-07-04

**Authors:** Jing Yan, Tomoaki Kahyo, Hengsen Zhang, Yashuang Ping, Chi Zhang, Shuyun Jiang, Qianqing Ji, Rafia Ferdous, Md. Shoriful Islam, Soho Oyama, Shuhei Aramaki, Tomohito Sato, Mst. Afsana Mimi, Md. Mahmudul Hasan, Mitsutoshi Setou

**Affiliations:** 1Department of Cellular and Molecular Anatomy, Hamamatsu University School of Medicine, 1-20-1 Handayama, Chuo-ku, Hamamatsu, Shizuoka 431-3192, Japan; yanjing20105@gmail.com (J.Y.); kahyo@hama-med.ac.jp (T.K.); hengsen19110@gmail.com (H.Z.); pingyashuang1989@gmail.com (Y.P.); zhangchi07.pegasus@gmail.com (C.Z.); shuyungoole@gmail.com (S.J.); jiqianqing@gmail.com (Q.J.); rafiaru22@gmail.com (R.F.); shoriful.pharmacy21@gmail.com (M.S.I.); oyamasoho@gmail.com (S.O.); aramaki.shh@gmail.com (S.A.); titon0620@gmail.com (T.S.); afsana.mimichem@gmail.com (M.A.M.); hasan.mahmudbio@gmail.com (M.M.H.); 2Quantum Imaging Laboratory, Division of Research and Development in Photonics Technology, Institute of Photonics Medicine, Hamamatsu University School of Medicine, 1-20-1 Handayama, Chuo-ku, Hamamatsu, Shizuoka 431-3192, Japan; 3Department of Orthopedic Surgery, Hamamatsu University School of Medicine, 1-20-1 Handayama, Chuo-ku, Hamamatsu, Shizuoka 431-3192, Japan; 4Department of Radiation Oncology, Hamamatsu University School of Medicine, 1-20-1 Handayama, Chuo-ku, Hamamatsu, Shizuoka 431-3192, Japan; 5Translational Biomedical Photonics, Institute of Photonics Medicine, Hamamatsu University School of Medicine, 1-20-1 Handayama, Chuo-ku, Hamamatsu, Shizuoka 431-3192, Japan; 6International Mass Imaging and Spatial Omics Center, Institute of Photonics Medicine, Hamamatsu University School of Medicine, 1-20-1 Handayama, Chuo-ku, Hamamatsu, Shizuoka 431-3192, Japan

**Keywords:** alpha-synuclein, ubiquitin-like 3, microsomal glutathione s-transferase 3, oxidative stress, small extracellular vesicles, neuroprotection, synucleinopathy

## Abstract

Aberrant aggregation of misfolded alpha-synuclein (α-syn), a major pathological hallmark of related neurodegenerative diseases such as Parkinson’s disease (PD), can translocate between cells. Ubiquitin-like 3 (UBL3) is a membrane-anchored ubiquitin-fold protein and post-translational modifier. UBL3 promotes protein sorting into small extracellular vesicles (sEVs) and thereby mediates intercellular communication. Our recent studies have shown that α-syn interacts with UBL3 and that this interaction is downregulated after silencing microsomal glutathione S-transferase 3 (MGST3). However, how MGST3 regulates the interaction of α-syn and UBL3 remains unclear. In the present study, we further explored this by overexpressing MGST3. In the split *Gaussia* luciferase complementation assay, we found that the interaction between α-syn and UBL3 was upregulated by MGST3. While Western blot and RT-qPCR analyses showed that silencing or overexpression of MGST3 did not significantly alter the expression of α-syn and UBL3, the immunocytochemical staining analysis indicated that MGST3 increased the co-localization of α-syn and UBL3. We suggested roles for the anti-oxidative stress function of MGST3 and found that the effect of MGST3 overexpression on the interaction between α-syn with UBL3 was significantly rescued under excess oxidative stress and promoted intracellular α-syn to extracellular transport. In conclusion, our results demonstrate that MGST3 upregulates the interaction between α-syn with UBL3 and promotes the interaction to translocate intracellular α-syn to the extracellular. Overall, our findings provide new insights and ideas for promoting the modulation of UBL3 as a therapeutic agent for the treatment of synucleinopathy-associated neurodegenerative diseases.

## 1. Introduction

Alpha-synuclein (α-syn) is a protein located primarily at the pre-synaptic terminus [1]. Abnormal aggregation of misfolded α-syn forms insoluble aggregates called Lewy bodies [2], which are a major pathological hallmark of related neurodegenerative diseases such as Parkinson’s disease (PD) [3]. Aggregated α-syn can be secreted by neurons via different pathways [4], not only through the non-canonical vesicle-mediated exocytosis pathway [5] but also released and delivered between neurons via tunneling nanotubes [6] and exosomes [7]. Potential anti-α-syn mechanisms include: inhibition of α-syn expression and aggregation; enhancement of α-syn degradation; and prevention of α-syn transmission [8]. Among them, the transition of α-syn from a physiological state to a pathological aggregation involves several factors, such as protein–protein interactions [9], PTM [10], oxidative stress [11], and gene mutations [12]. Therefore, the development of disease-modifying drugs targeting insoluble aggregates and their constituent molecules is of great interest.

Ubiquitin-like 3 (UBL3) is a membrane-anchored ubiquitin-fold protein [13], the protein that post-translationally modifies (PTM) substrate proteins and anchors to the plasma membrane or early endosomes [14], which promotes protein sorting to small extracellular vesicles (sEVs) [15] and thus mediates intercellular communication [16]. sEVs are nanometer-sized secreted membrane vesicles that are derived from various cell types through multivesicular bodies [17]. The biosynthesis of exosomes involves their origin in endosomes, and thus, it contains different components, including RNA (mRNA, miRNA, lncRNA, and rRNA), lipids, DNA, proteins, and metabolites [17]. Previous studies have identified 1241 proteins that can interact with UBL3, including more than 22 disease-associated proteins, including molecules associated with neurodegenerative diseases [16]. In our previous report, we demonstrated that α-syn and UBL3 interact [18] and that silencing of microsomal glutathione s-transferase 3 (MGST3) downregulates this interaction [19].

Glutathione transferase (GST) isozymes are encoded by three separate gene families (known as cytoplasmic, microsomal, and mitochondrial transferase) [20], where the majority of the microsomal GSTs are involved in the biotransformation of arachidonic acid metabolites and are therefore known as the membrane-associated proteins in eicosapentaenoic acid and glutathione metabolism (MAPEG) family [21]. And MGST3 encodes one of the MAPEG family members. It acts as an enzyme that promotes the production of the lipid inflammatory mediator leukotriene C [22] and also exhibits glutathione-dependent peroxidase activity against lipid hydroperoxides [23]. In the brain, the high level of MGST3 mRNA expression contrasts with that of other members of the family [24]. Recently, it has been shown that *MGST3* is co-expressed with genes associated with hippocampal size reduction in neurodegenerative diseases: Huntington’s disease, Alzheimer’s disease, and Parkinson’s disease [25]. Therefore, we conducted further studies on the effect of MGST3 on the interaction of α-syn with UBL3.

We hypothesize that MGST3 acts as a modulator to regulate the interaction of α-syn with UBL3 and mediates the sorting of α-syn into small extracellular vesicles, which would provide a new idea to remove the intracellular α-syn accumulation and thus provide a treatment for synucleinopathies. In the present study, we aimed to investigate whether MGST3 could act as a regulator of α-syn-UBL3 interaction and attempted to explore the function of MGST3 in regulating this interaction.

## 2. Result

### 2.1. MGST3 Affects α-Syn and UBL3 Interaction by Split Gaussia Luciferase Complementation Assay

In our previous study, we found that silencing of MGST3 significantly reduced the interaction between α-syn and UBL3 using split *Gaussia* complementary luciferase analysis [19]. To further explore the effect of MGST3 on the interaction between α-syn and UBL3, we both silenced and overexpressed MGST3 in the HEK293 cells. Split *Gaussia* complementary luciferase assay of cell culture and cell lysates collected 72 h after transfection showed that the luminescence intensity in the cell culture medium (Figure 1A) and cell lysate (Figure 1B) was significantly reduced after silencing MGST3, which was consistent with our previous results. The luminescence intensity in the cell culture medium and the lysate was significantly enhanced after overexpression of MGST3. These results show that MGST3 can affect α-syn and UBL3 interactions in HEK293 cells.

### 2.2. Silencing or Overexpression of MGST3 Has Not Significantly Altered the Expression of α-Syn and UBL3

To further explore how MGST3 altered the interaction between α-syn and UBL3 after co-transfection of siRNA, NGluc-UBL3, α-syn-CGluc, and MGST3-HA into HEK293 cells, proteins in cell lysates were analyzed by Western blotting. The endogenous MGST3 level was actually reduced after MGST3 silencing, and the overexpressed MGST3-HA (1102.15 Da) was confirmed (Figure 2A). There was a weak effect on the protein level of UBL3, whereas there was no significant change in the α-syn protein level. Next, we investigated the effect of MGST3 on α-syn and UBL3 gene expression by RT-qPCR (Figure 2B). The relative expression of MGST3 was significantly altered by silencing and overexpression compared to the control groups. Consistent with the protein blotting results, the expression of α-syn and UBL3 was not affected when MGST3 was either silenced or overexpressed (Figure 2C,D).

### 2.3. MGST3 Can Upregulate the Co-Localization of α-Syn and UBL3 in HEK293 Cells

These results indicate that overexpression of MGST3 could increase the localization of α-syn and UBL3. First, we constructed UBL3 tagged with mStayGold, which is a bright and highly photostable fluorescent protein. Then, co-transfection with α-syn, mStayGold-UBL3, and MGST3 was conducted in HEK293 cells for immunocytochemical staining. The results indicated that the co-localization of α-syn and UBL3 was enhanced after overexpression of MGST3 (Figure 3A,B).

### 2.4. MGST3 Was Able to Attenuate the Effect of Oxidative Stress on the Interaction between α-Syn and UBL3

MGST3 has a glutathione-dependent peroxidase activity of lipid hydroperoxides [26,27], and we previously reported that 800 µM of H_2_O_2_ mimicked oxidative stress and downregulated the interaction between α-syn and UBL3 [19]. To verify whether the effect of MGST3 overexpression on the interaction between α-syn and UBL3 is altered under the excess oxidative stress, we treated HEK293 cells co-transfected with siRNA, α-syn-CGluc, NGluc-UBL3, and MGST3 with 800 µM H_2_O_2_. Cell culture media from different experimental groups were collected for luminescence assay, and we assessed the cell activity using MTT. The luminescence intensity was significantly reduced in the cell culture medium under excess oxidative stress (Figure 4A). On the other hand, the luminescence intensity increased in the overexpression of MGST3, whether with or without oxidative stress treatment. Moreover, we observed that oxidative stress significantly reduced cell viability (Figure 4B), but there was no significant difference in cell activity between the groups under the same transfection conditions. To exclude that there were differences in cell viability by H_2_O_2_ on the different conditioned groups that affected the luminescence intensity, we divided the luminescence intensity by the cell viability to assess the relative luminescence values (Figure 4C). It was found that while H_2_O_2_ downregulated the interaction between α-syn and UBL3, overexpression of MGST3 was able to significantly rescue the effect of oxidative stress on their interaction.

### 2.5. Overexpression of MGST3 Increases α-Syn Secretion into the Extracellular during Oxidative Stress

To investigate whether the interaction between α-syn and UBL3 affects intracellular and extracellular α-syn distribution under different expression levels of MGST3 and oxidative stress conditions, we analyzed the α-syn-HiBiT distribution in cell lysate and culture medium using HiBiT bioluminescence assay in various experimental groups. Figure 5A–C shows that α-syn was mainly distributed in the intracellular, and silencing MGST3 increased intracellular α-syn both under normal and oxidative stress conditions, while in oxidative stress conditions, overexpression of MGST3 decreased α-syn in the cell lysate and increased α-syn in the culture medium, meaning that intracellular α-syn was only 0.502-fold that in the culture medium (Figure 5D). In Western blotting, we also found that overexpression of MGST3 under oxidative stress increased α-syn in the culture medium and decreased α-syn in the cell lysate (Figure 5E), compared with the control. Thus, overexpression of MGST3 under oxidative stress promoted the secretion of α-syn to the outside of the cell.

## 3. Discussion

In this study, we investigated the effects of silencing and overexpression of MGST3 on the interaction between α-syn and UBL3 in cells. We found that MGST3 upregulated the interaction between α-syn and UBL3 with no significant change in the expression of α-syn and UBL3 in cells. In addition, MGST3 was capable of rescuing the effect of oxidative stress on the interaction between α-syn and UBL3 to a certain extent.

The interaction between MGST3 on α-syn and UBL3 was observed to be positively correlated by split *Gaussia* luciferase complementation assay (Figure 1). As detected by proteomic analysis, in the protein interactome dependent on the binding of two cysteine residues at the C-terminus of UBL3 via disulfide bonds to modify its protein interactome [16], we found that MGST3 was able to influence α-syn and UBL3 interactions. So, it is also possible that this interaction result is interfered with by MGST3, affecting UBL3 or α-syn expression. Therefore, we quantitated the proteins to detect the protein level expression and showed that the protein expression of α-syn was not significantly altered by the silencing and overexpression of MGST3, but the UBL3 protein level appeared to be weakly affected by MGST3 (Figure 2A). However, analysis at the level of expression of mRNA genes indicated that UBL3 and α-syn were not affected by MGST3 (Figure 2C,D). Reduced glutathione (GSH) plays an important role as the major non-protein sulfhydryl group in the cell. In vivo, it is mediated by GST to form glutathione complex (GS-x) in a binding reaction and glutathione peroxidase (GP-x) to glutathione disulfide (GSSG) [28]. The thiol portion of the cysteine residue of GSH is thus important in resistance to oxidative defense [29], xenobiotics [30], gene expression [31], eicosanoid metabolism, and regulation of the cell cycle [28]. And the oxidative modification of cysteine residues is a major PTM participating in the ROS-mediated cell signaling pathway [32]. Therefore, we hypothesized that MGST3 regulates UBL3 by covalently modifying thiols on UBL3 cysteine residues and thereby affects the interaction of α-syn with UBL3. However, there is no study evidence for a molecular relationship between MGST3 and UBL3, and we need more investigations for elucidation in the future.

In previous studies, we found that silencing of MGST3 was able to reduce the co-localization of UBL3 with α-syn proteins, whereas UBL3 was shown to be predominantly distributed as a membrane-anchored protein in the cell membrane. Subsequently, we constructed mStayGold-UBL3 that stably expresses fluorescence to reconfirm this fact (Figure 3). We found that MGST3 did not alter the expression of α-syn and UBL3 at the mRNA level, but it slightly affected the expression of UBL3 at the protein level, thus affecting the interaction between α-syn and UBL3 (Figure 2). In co-localization analysis with α-syn, overexpression of MGST3 enhanced the co-localization of α-syn with UBL3, and in combination with previous results that silencing of MGST3 downregulated this co-localization, this cellular staining further demonstrated that MGST3 acts as a regulatory factor that affects the interaction between α-syn and UBL3. However, the functional role of MGST3 in effecting the interaction between α-syn and UBL3 remains unclear.

It has also been noted that MGST3 expression correlates with the distribution of excitatory glutamatergic neurons (major neurons formed in the hippocampus, neurons in the cortex and thalamus, and DRG neurons) and that the excitotoxic effect of glutamate overload is a mechanism of neurological injury, which is consistent with the detoxifying and neuroprotective role of MGST3 in oxidative stress in the rat nervous system [24]; that the expression of the MGST3 gene correlates with the size of the hippocampus correlates with hippocampal size, and its dysregulation leads to neurodegenerative diseases [25]. Pleiotrophin deletion upregulates caspase 6, which is associated with axonal degeneration and neurodegenerative diseases, and the deleterious effects produced can be partially compensated for by the simultaneous induction of the neuroprotective gene MGST3 [33]. Mitochondrial ROS derived from astrocytes exhibit a physiological brain-protective function through the activation of NRF2 [34], which can dependently regulate MGST3 to protect cells from iron death [35]. Since cellular homeostasis is an important process for protein function [36], over-oxidative stress is one of the important molecular mechanisms that leads to neurodegeneration, as well as protein aggregation and mitochondrial dysfunction [37]. MGST3 is known to have glutathione-dependent peroxidase activity that protects cells against oxidative damage [26], and it is also used as a biomarker of oxidative stress [38]. The effect of oxidative stress on the interaction between α-syn and UBL3 was explored in a previous study, showing that oxidative stress downregulated this interaction; however, silencing of MGST3 did not further reduce this interaction, thought to be possibly due to the intensity of oxidative stress being beyond the regulatory range of endogenous MGST3 [19]. It is noteworthy that neither silencing nor overexpression of MGST3 affected cell viability in the normal cell growth condition, and overexpression of MGST3 under oxidative stress conditions did not repair the disrupted cell viability. However, after oxidative stress treatment, the overexpression of MGST3, while not fully repairing the effect of oxidative stress on the interaction between α-syn and UBL3, reversed to an extent the downregulation of this impact by oxidative stress (Figure 4C). Our results demonstrate that MGST3 stabilizes protein–protein interactions under oxidative stress conditions.

In our study, we found that silencing MGST3 hindered α-syn transport to the extracellular (Figure 5A,C); overexpression of MGST3 was able to enhance α-syn interaction with UBL3 (Figure 1A,B and Figure 4C) and cellular co-localization (Figure 3A,B) without significantly altering α-syn in the cell (Figure 2A,D and Figure 5B,E). However, when exposed to oxidative stress conditions, overexpression of MGST3 increased α-syn interaction with UBL3 (Figure 4C) while at the same time promoting intracellular α-syn transport to the outside of the cell (Figure 5D,E). When neurons are subjected to cellular stress or pathological injury, α-syn can self-protect through small extracellular vesicle secretion in neurons [39]. UBL3 acts as a post-translational modifier to promote protein sorting into small extracellular vesicles [16], and MGST3 can rescue the effect of oxidative stress conditions on the interaction between α-syn and UBL3, thereby reducing the aggregation of α-syn in the cell. This evidence provides a new therapeutic target for MGST3 as a molecule that regulates the interaction of α-syn with UBL3 in neurodegenerative diseases.

Since there are limitations in the current study, we only illustrated that MGST3 can act as a modulator to affect α-syn-UBL3 interaction, and the molecular mechanism of its existence is still not completely clear to us. In the future, we need to observe the interrelationships of MGST3 on α-syn and UBL3 proteins between each other and with exosomes by immunoprecipitation and immunocytochemical staining. In this way, we will be able to fully interpret the effect of MGST3 on the interaction between α-syn and UBL3, providing reliability and therapeutic prospects for the treatment of neurodegenerative diseases.

## 4. Materials and Methods

### 4.1. Plasmid and siRNA

We used the conventional molecular biology techniques and PCR to insert mStayGold [40] (primer sequences: forward: 5′-CGCGGATCCATGGTGTGTCTACAGGCGAGGAG-3′; reverse: 5′-CTAGCTAGCCAGGTGGGGCCTCCAGG-3′) into the UBL3 (NM_007106) expressing pcDNA3 vector, located between the BamHI and NheI sites before the UBL3 sequence. NGluc-UBL3, α-syn-CGluc, 3xFlag-UBL3, 6xMYC-α-syn, and Gluc plasmids were used previously in our laboratory [18]. MGST3_OHu11138C_pcDNA3.1(+)-C-HA (MGST3-HA) plasmid cloning strategy: EcoRI/XhoI; 3_SNCA-HibiT_pcDNA3.1(+) (α-syn-HiBiT) plasmid cloning strategy: KpnI/XhoI were customized by Genscript USA Inc. The MGST3 siRNA (s8762) Sense 5′- 3′ AGA ACA CGU UGG AAG UGU Att Antisense 5′- 3′ UAC ACU UCC AAC GUG UUC Ugg and siRNA negative control #1 were purchased from Silencer Select (Ambion, Life Technologies, Carlsbad, CA, USA).

### 4.2. Cell Culture and Transfection

Human embryonic kidney (HEK293) cells (RIKEN Cell Bank, Ibaraki, Japan) were cultured in Dulbecco’s modified Eagle medium (DMEM, GIBCO, 11965-092) containing 10% fetal bovine serum (FBS) (Sigma-Aldrich, St. Louis, MI, USA). Cultures were incubated at 37 °C in a 5% CO_2_ humidified incubator (WY-320, Thermo Scientific, Waltham, MA, USA). Cells were cultured in culture dishes to 80–90% confluency and were transiently transfected with cDNA plasmid using Lipofectamine 2000 Transfection Reagent (Thermo Fisher Scientific, Waltham, MA, USA) diluted in Opti-MEM reduced serum medium (Thermo Fisher Scientific, Waltham, MA, USA) according to the reagent instructions, which have been described previously [19].

### 4.3. Luciferase Assay

Cells were incubated for 72 h after transfection. The cell culture medium was collected and centrifuged at 1200 rpm for 5 min to remove cell debris. The supernatant was then added to 17 µg/mL coelenterazine (Cosmo Bio, Kyodo, Japan) in Opti-MEM, and luminescence was immediately measured using a BioTek Synergy H1 microplate reader (Agilent, Santa Clara, CA, USA).

### 4.4. Western Blotting (WB)

Cell cultures were washed with ice-cold PBS, and cells were harvested and pelleted by centrifugation at 2000× *g* for 5 min at 4 °C. Cell pellet was resuspended and lysed with 1% Triton lysate buffer (50 mM Tris-HCl [pH 7.4], 100 mM NaCl, and 1% [v/v] Triton X-100) for 30 min at 4 °C. Cell lysates were centrifuged at 20,000× *g* for 15 min at 4 °C to remove cell debris and unlysed cells. Quantification of protein concentration was performed using the Pierce BCA Protein Assay Kit (23227, Thermo Fisher Scientific, Hanover Park, IL, USA). For the WB analysis, which has been described previously [19], 10 µg of total proteins were loaded after being treated with 2-mercaptoethanol (βME+) 2× sodium dodecyl sulfate (SDS) sample loading buffer (100 mM Tris-HCl [pH 6.8], 4% SDS, 20% glycerol, and 0.01% bromophenol blue) at 95 °C for 5 min, separated on SDS-PAGE gels and transferred to polyvinylidene fluoride (PVDF) membranes. The membranes were blocked with 5% skim milk shake for 1 h at room temperature, then incubated overnight at 4 °C with the indicated primary antibodies. The membranes obtained by WB were immunoblotted using the following antibodies: anti-MGST3 antibody (Abcam, ab192254; 1: 1000), anti-UBL3 antibody (ABclonal, A4028, 1:1000 dilution), anti-HA antibody (Roche, 3F10, 1:1000 dilution), anti-α-syn antibody (BioLegend, 834304, 1:1000 dilution), anti-HiBiT antibody (Promega, N7200, 1:1000 dilution), and anti-β-actin antibody (Cell Signaling, 3700, 1:1000 dilution). Primary antibodies were prepared in Tween-20 (+) Tris Buffered Saline (TBS-T; 100 mM Tris-HCl [pH 8.0], 150 mM NaCl, 0.5% [v/v] Tween-20). The membrane was washed thrice with TBS-T and then incubated with horseradish peroxidase (HRP)-conjugated anti-rabbit secondary antibody (Cell signaling, 1:5000 dilution) prepared in blocking buffer for 1 h at room temperature shaken. After being washed in TBS-T, the membranes were followed by immunoreactive protein developed using the Enhanced Chemiluminescence Kit (32106, Thermo Fisher Scientific, Waltham, MA, USA) with the FUSION FX imaging system (Vilber Lourmat, Collégien, Seine-et-Marne, France) for detection. ImageJ 2.0 software was used for analysis, and the uncropped version of each image is shown in Appendix A.

### 4.5. RT-qPCR

Cells were transfected for 48 h, and total RNA was extracted according to the manufacturer’s instructions for the RNeasy mini kit (#74104, QIAGEN, Hilden, Germany). The extracted RNA was reverse transcribed to cDNA using the Rever Tra Ace qPCR RT kit (FSQ-101, TOYOBO, Tokyo, Japan). RT-qPCR was then performed on a QuantStudio 3 real-time PCR system (Thermo Fisher Scientific, Waltham, MA, USA) using the THUNDERBIRD SYBR qPCR Mix (#QPS-201, TOYOBO, Tokyo, Japan) following standard reaction conditions. The following primers were used and customized from Integrated DNA Technologies, Inc.: MGST3 (forward: AGAACCCAGCAAGCGTAGTC, reverse: GCCCAAGCCACTTTTAACCC), UBL3 (forward: ATGTCCAGTAATGTCCCGGC, reverse: GACTGCTGACCTGCTCTTCTT), α-syn (forward: GTGGCTGCTGCTGCTGAGAAAAC, reverse: CACCACTGCTCCTCCAACAT), and β-actin (forward: TCACCATGGATGATGATATATCGC, reverse: ATAGGAATCCTTCTGACCCATGC). The relative expression of target RNAs was calculated by the 2^−ΔΔC^ method and normalized by the level of the housekeeping gene β-actin.

### 4.6. Immunocytochemistry

HEK293 cells were co-transfected with mStayGold-UBL3, 6xMYC-α-syn, and MGST3 using Lipofectamine 2000 and Poly-L-lysine-coated cover glasses [41] on the bottom of cell culture dishes. After 24 h, cells were fixed with methanol for 5 min; blocked with 1% BSA/PBS for 1 h; incubated with primary antibodies and anti-MYC antibodies (MBL, M1923, 1:500 dilution, Woods Hole, MA, USA) for 18 h at 4 °C and secondary antibodies Alexa Fluor 647, 1:500 dilution, Carlsbad, CA, USA) for 1 h; and mounted with VECTASHIELD Mounting Medium (Vector), which has been described previously [19]. Confocal images were acquired with a 63× objective lens on a confocal laser microscope (Leica TCS SP8). The co-localizations of α-syn with UBL3 were analyzed using ImageJ 2.0 software.

### 4.7. Oxidative Stress

To investigate the effect on α-syn and UBL3 by varying concentrations of hydrogen peroxide (H_2_O_2_) and silencing MGST3 based on oxidative stress, we measured the relative luminescence intensity of α-syn and UBL3 in live cells, which has been described previously [19]. Cells were transduced into 96-well plates with 100 μL per well of DMEM (10% FBS) and incubated overnight at 37 °C in 5% CO_2_ humidified incubator. The next day, H_2_O_2_ was diluted with pre-warmed DMEM (10% FBS), and the old medium of 50 μL/well was replaced with 100 μL of DMEM (10% FBS) with various final concentrations of H_2_O_2_ (0 µM, 100 µM, 200 µM, 400 µM, and 800 µM). Cells were further incubated at 37 °C for 48 h, and then CM was collected for luminescence intensity assay. For analysis of cell viability by the 3-(4,5-dimethylthiazol-2-yl)-2,5-diphenyl-2H-tetrazolium Bromide (MTT) cell growth kit (CT02, Millipore, MA, USA); after collection of cell culture medium for luminescence intensity assay, fresh pre-warmed DMEM (10% FBS) medium was added to 100 μL per well with 10 μL of MTT reagent and incubated in the incubator at 37 °C for 4 h. Subsequently, 100 μL of solubilization buffer (isopropanol with 0.04 N hydrogen chloride) was added to each well, and the absorbance value at OD450 nm of each well was measured with the microplate reader.

### 4.8. HiBiT Bioluminescence Assay

To explore the intracellular and extracellular distribution of α-syn, the siRNA, 3xFlag-UBL3, α-syn-HiBiT, and MGST3-HA were co-transfected into HEK293 cells. Oxidative stress treatment (final concentrations of H_2_O_2_ is 800 µM) was as previously described conditions. Cells were further incubated at 37 °C for 48 h, the culture media (CM) were collected and centrifuged at 1200 rpm for 5 min, and 100 µL of supernatant were transferred into black glass bottom 96-well cell culture plates (61-9713-47 EZVIEW). The cell lysates were prepared by adding 100 µL of DMEM (10% FBS) into each well. The confluent cells were detached from the surface and lysed with NanoGlo HiBiT Lytic Reagent (Promega N3030, a buffer containing the recombinant N-terminus of nanoluciferase (LgBiT) and nanoluciferase substrate Furimazine). The luminescence intensity of untreated DMEM (10% FBS) was set as a background. The luminescence intensity of α-syn-HiBiT was measured in the plate by adding the same volume of NanoGlo HiBiT Lytic Reagent to each well. The reagent-treated cell lysates and culture media were incubated for 10 min rotation. Then, the luminescence intensity of cell lysate and culture media was measured with a microplate reader (BioTek, Winooski, VT, USA) at a one-second integration time. By deducting the background’s luminescence intensity, all of the cell lysate and culture media’s luminescence intensities were adjusted.

### 4.9. Statistical Analysis

Statistical analysis was performed using GraphPad Prism 6 software. Results from at least three independent experiments are presented as the mean ± S.D. Statistical significance was assessed by two-tailed Student’s t-test for two groups and one-way analysis of variance (ANOVA) for more than two groups. *p* < 0.05 was considered statistically significant.

## 5. Conclusions

Our results show that silencing of MGST3 inhibits α-syn and UBL3 interaction; in contrast, this interaction is enhanced by overexpression of MGST3. Under oxidative stress conditions, MGST3 was able to rescue the inhibition of α-syn-UBL3 interaction by oxidative stress and promote intracellular α-syn translocation to the outside of the cell. Overall, our findings provide new insights and ideas for promoting the modulation of UBL3 as a therapeutic agent for the treatment of synucleinopathy-associated neurodegenerative diseases.

## Figures and Tables

**Figure 1 ijms-25-07353-f001:**
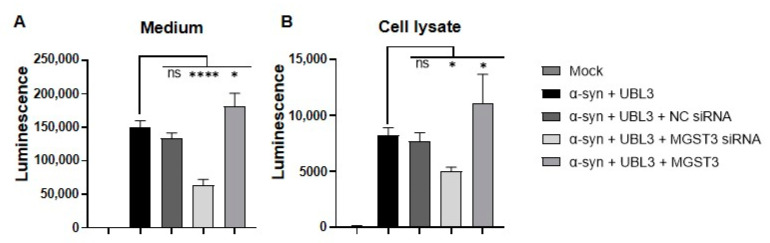
Effect of MGST3 on the interaction of α-syn with UBL3. Luminescence intensity in culture medium and cell lysate were evaluated by the *Gaussia* luciferase complementation assay after co-transfection of siRNA, NGluc-UBL3, α-syn-CGluc, and MGST3 into HEK293 cells. The luminescence ± SD in triplicate experiments is shown. One-way ANOVA and Dunnett’s post hoc test were performed. ns: non-significant; *: *p* < 0.05; ****: *p* < 0.0001; NC siRNA: negative-control siRNA.

**Figure 2 ijms-25-07353-f002:**
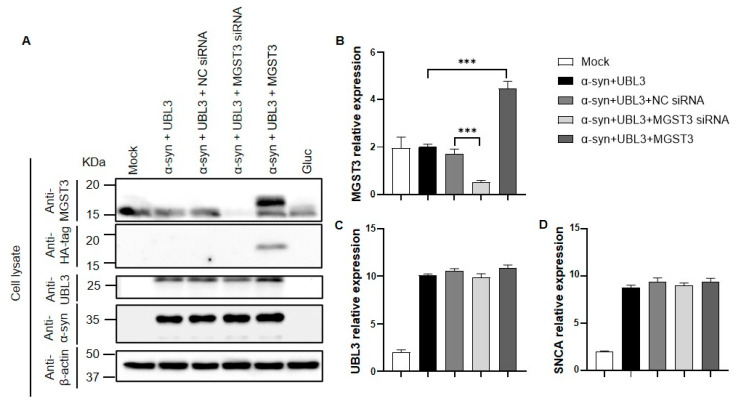
Effect of MGST3 in α-syn and UBL3. (**A**) Effect of MGST3 in α-syn and UBL3 at the protein level. The siRNA, NGluc-UBL3, α-syn-CGluc, and MGST3-HA were co-transfected into HEK293 cells, and cell lysate was blotted with various antibodies. (**B**–**D**) Effect of MGST3 on α-syn and UBL3 gene expression. The siRNA, UBL3, α-syn, and MGST3 were co-transfected into HEK293 cells, and the expression of MGST3, UBL3, and α-syn in the cells. The gene expression of MGST3, UBL3, and SNCA were analyzed for 48 h after transfection by RT-qPCR, respectively. Unpaired t-test (*n* = 3 per group); all data represented as mean ± SD. ***: *p*-value < 0.005; NC siRNA: negative-control siRNA.

**Figure 3 ijms-25-07353-f003:**
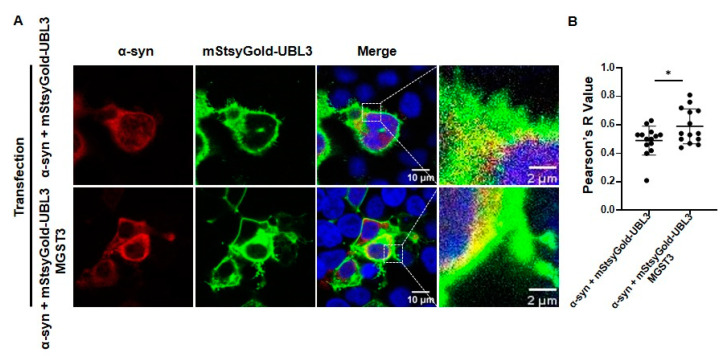
Effect of MGST3 on co-localization of α-syn and UBL3. (**A**) Representative images of the effect of MGST3 on the co-localization of α-syn and UBL3 were observed after co-transfection of α-syn and UBL3, silencing or overexpression of MGST3 in HEK293 cells. Red represents α-syn, green represents UBL3, blue represents the cell nucleus, and yellow shows the co-localization of α-syn and UBL3. The regions in the dotted box are shown as a single confocal image in the inset. Scale bars, 10 and 2 μm. (**B**) Quantitative analysis of the effect of overexpression of MGST3 on the co-localization of mStayGold-UBL3 with 6xMYC-α-syn. Unpaired t-test (*n* = 14 per group); all data represented as mean ± SD. *: *p*-value < 0.05.

**Figure 4 ijms-25-07353-f004:**
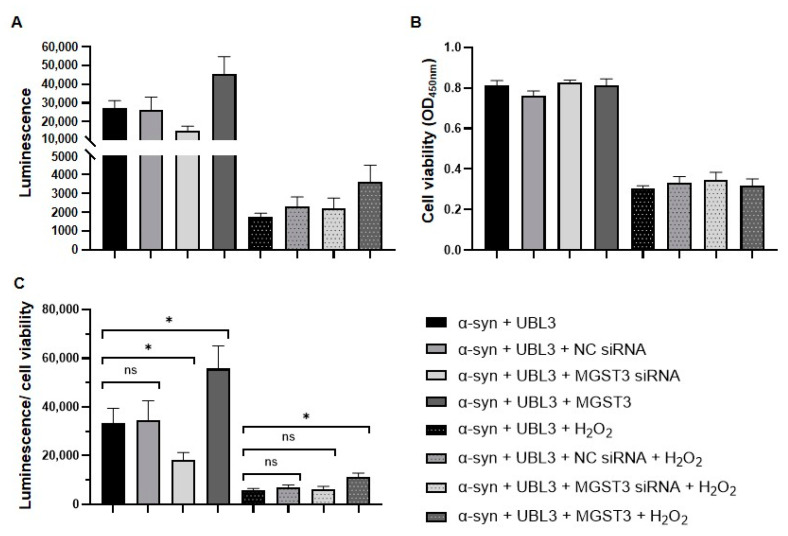
Effect of MGST3 on α-syn and UBL3 interactions after treatment with H_2_O_2_. (**A**) The luminescence intensity of cell cultures was measured after co-transfection of α-syn-CGluc and NGluc-UBL3 into HEK293 cells, in which the H_2_O_2_ treatment concentration was 800 µM. (**B**) Cell viability of various experimental groups of HEK293 cells. (**C**) Luminescence-to-cell viability ratio of HEK293 cell cultures co-transfected with α-syn-CGluc and NGluc-UBL3. The luminescence ± SD, cell viability ± SD, and ratio ± SD in triplicate are shown. One-way ANOVA and Dunnett’s post hoc test were performed. ns: non-significant; *: *p* < 0.05.

**Figure 5 ijms-25-07353-f005:**
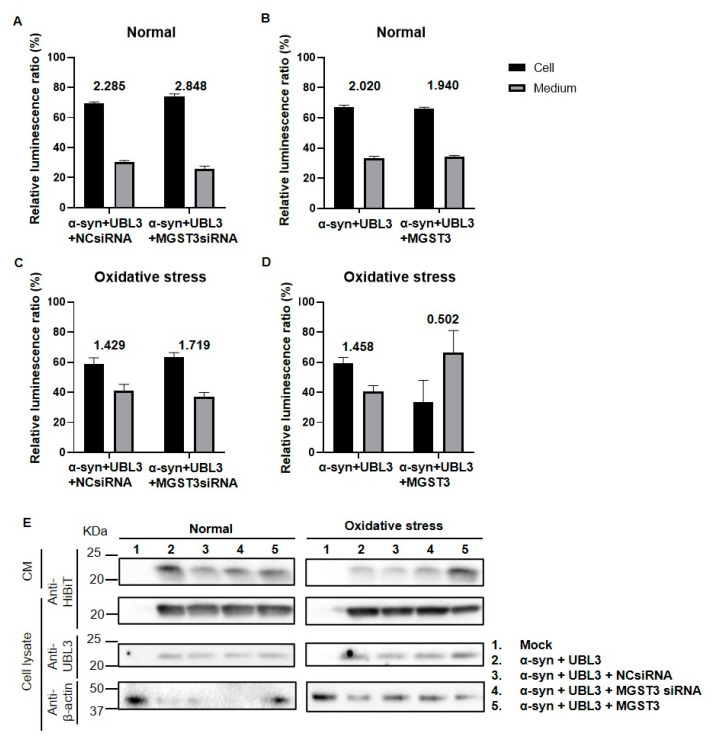
Overexpression of MGST3 increases α-syn secretion into the extracellular during oxidative stress. The siRNA, 3xFlag-UBL3, α-syn-HiBiT, and MGST3-HA were co-transfected into HEK293 cells. (**A**–**D**) show the relative luminescence ratios of α-syn-HiBiT detected in the cell and culture medium under MGST3 knockdown or overexpression and oxidative stress conditions. The values in (**A**) 2.285, 2.848, (**B**) 2.020, 1.940, (**C**) 1.429, 1.719, and (**D**) 1.458, 0.502 are the cell lysate-to-culture medium ratios of α-syn-HiBiT in each experimental group. All data are represented as mean ± SD. (*n* = 6 in the Normal group, *n* = 5 in the Oxidative stress group.) (**E**) Cell lysate and culture medium were blotted with various antibodies.

## Data Availability

Data are contained within the article or Appendix A.

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
