# Peer review of "Alpha-Synuclein Interaction with UBL3 Is Upregulated by Microsomal Glutathione S-Transferase 3, Leading to Increased Extracellular Transport of the Alpha-Synuclein under Oxidative Stress"

_ijms, 2024, doi:10.3390/ijms25137353_

Round 1

Reviewer 1 Report

Comments and Suggestions for Authors

This manuscript probes the effects of microsomal glutathione s-transferase (MGST3) on the potential interactions between alpha synuclein and ubiquitin-like 3 (UBL3). Such interactions potentially lead to the extracellular transport of alpha synuclein out of the cell, and as such, is a potential target for medical applications.

The authors' results seem to demonstrate clearly that MGST3 expression modulates the interactions between alpha synuclein and UBL3, and that this modulation is in fact achieved at the level of protein-protein interactions, since expression of these two proteins were not affected by the MGST3 expression level. It is unfortunate, however, that the authors do not provide data that probes for the downstream consequences of these interactions, specifically, if the interactions change in any way the post-translational modification pattern of alpha synuclein, or result in any changes in localization of alpha synuclein into the various locations in the cell that lead to transport out of the cell. The data provided is a very strong base from which the authors may extend their research in many ways, and it is a shame that they stop at discussing these potential routes in their discussion. As a result, the manuscript gives off the impression of a preliminary mechanistic probe, and I fear will be of limited interest to other researchers.

A suggestion, if possible, would be to try and isolate any post translational modifications on the alpha-synuclein protein at various MGST3 expression levels and oxidative stress conditions, which might give additional details to the events that follow this interesting interaction.

Other points; I was unable to deduce the significance of the additional blot images that were provided by the authors as non-published additional data. Are there any additional explanational texts that I was not given access to regarding these materials? As a result I have not incorporated these data in preparing my review.

Comments on the Quality of English Language

Regarding the English usage in the manuscript, I felt that some very minor changes, particularly in the Materials section, would clarify some of the details of the study. However, the overall English is very good, so this comment is an optional one to the authors.

Author Response

Dear Reviewer,

Thank you very much for your valuable comments and suggestions. You have been a great help in enhancing our manuscript with your enlightened scientific thinking. Attached is my response to your review. I hope it will be to your satisfaction.

Reviewer 2 Report

Comments and Suggestions for Authors

Title: Alpha-synuclein interaction with UBL3 is upregulated by microsomal glutathione S-transferase 3.

Summary

In this manuscript, Jing Yan et al. studied how the silencing or overexpression of MGST3 (microsomal glutathione S-transferase 3) affects the interactions between alpha-synuclein and UBL3 (ubiquitin-like 3). The data showed that knockdown of MGST3 inhibited the interactions between α-syn and UBL3, and overexpression of MGST3 enhanced their interactions. The oxidative stress impaired their interactions, while the overexpression of MGST3 attenuated this effect. This manuscript is not publishable in the International Journal of Molecular Sciences.

Major and minor comments are listed below:

1. It seems that the research data shown in this manuscript is a continuation of the experimental results presented in the ref [19]. The current data lacks novelty and independence.

2. The data can not support the major conclusions of this paper. More experiments are needed to consolidate. For example, immunoprecipitation experiments are needed to confirm their interactions, or another independent technique is needed to validate the interactions between α-syn and UBL3, besides the luciferase assay.

3. In the Results, the subtitles of these two sections “1. MGST3 affects α-syn and UBL3 interaction by split Gaussian luciferase complementation assay.” and 2. Silencing or overexpression of MGST3 on α-syn and UBL3 expression.” are confusing. Try to rewrite these two subtitles.

4. -Figure 3, what is Pearson’s R value? How is it calculated?

5. The image in the Conclusion should be deleted.

6. What are a-syn-CGluc and MYC-a-syn constructs? The molecular weight of a-syn is about 15kDa, while the WB membrane shows a much higher molecular weight (~35kDa).

Comments on the Quality of English Language

Minor editing of English language required.

Author Response

Dear Reviewer,

Thank you very much for providing your valuable comments and suggestions. We have enhanced the manuscript based on your suggestions. Attached is my response to your review. I hope our response will be satisfactory to you.

Round 2

Reviewer 1 Report

Comments and Suggestions for Authors

The additional experiments performed by the authors are very interesting, and I believe have raised the significance of the submission.

Minor English editing should be attempted in the revised manuscript; for example, the title might be modified as "Alpha synuclein interaction with UBL3 is upregulated by microsomal glutathione S-transferase-3, leading to increased extracellular transport of the former under oxidative stress". 

Comments on the Quality of English Language

Minor English editing should be attempted in the revised manuscript; for example, the title might be modified as "Alpha synuclein interaction with UBL3 is upregulated by microsomal glutathione S-transferase-3, leading to increased extracellular transport of the former under oxidative stress". 

Author Response

Comments 1: Minor English editing should be attempted in the revised manuscript; for example, the title might be modified as "Alpha synuclein interaction with UBL3 is upregulated by microsomal glutathione S-transferase-3, leading to increased extracellular transport of the former under oxidative stress".

Response 1: Thank you very much for your valuable comments on our manuscript with great enhancement. 
1.    We have revised the title of the manuscript with reference to your comments to: “Alpha-synuclein interaction with UBL3 is upregulated by microsomal glutathione S-transferase 3, leading to increased extracellular transport of the alpha-synuclein under oxidative stress.
2.    We checked the manuscript and added information about the antibodies used in lines 183-185 of the Materials and Methods.
3.    We have revised lines 457-464 to improve the Conclusion.